# BRAINCODE for Cognitive Impairment Diagnosis in Older Adults: Designing a Case–Control Pilot Study

**DOI:** 10.3390/ijerph19095768

**Published:** 2022-05-09

**Authors:** Pedro Rocha, Paulina Clara Dagnino, Ronan O’Sullivan, Aureli Soria-Frisch, Constança Paúl

**Affiliations:** 1Departamento de Ciências do Comportamento, ICBAS—Instituto de Ciências Biomédicas Abel Salazar, Universidade do Porto, Rua Jorge de Viterbo Ferreira, 228, 4050-313 Porto, Portugal; paul@icbas.up.pt; 2CINTESIS—Centro de Investigação em Tecnologias e Serviços de Saúde, R. Dr. Plácido da Costa, 4200-450 Porto, Portugal; 3Starlab Barcelona SL, Neuroscience Business Unit, Avda. Tibidabo 47 bis, 08035 Barcelona, Spain; paulinaclara.dagnino@starlab.es (P.C.D.); aureli.soria-frisch@starlab.es (A.S.-F.); 4Centre for Gerontology and Rehabilitation, School of Medicine, University College Cork, College Road, T12 K8AF Cork, Ireland; ronan.osullivan@ucc.ie

**Keywords:** healthy ageing, neurocognitive disorders, dementia, mild cognitive impairment, EEG diagnosis, medical devices, digital health

## Abstract

An early, extensive, accurate, and cost-effective clinical diagnosis of neurocognitive disorders will have advantages for older people and their families, but also for the health and care systems sustainability and performance. BRAINCODE is a technology that assesses cognitive impairment in older people, differentiating normal from pathologic brain condition, based in an EEG biomarkers evaluation. This paper will address BRAINCODE’s pilot design, which intends to validate its efficacy, to provide guidelines for future studies and to allow its integration on the SHAPES platform. It is expected that BRAINCODE confirms a regular clinical diagnosis and neuropsychologic tests to discriminate ‘normal’ from pathologic cognitive decline and differentiates mild cognitive impairment from dementia in older adults with/without subjective cognitive complains.

## 1. Neurocognitive Disorders in a Decade of Healthy Ageing

The United Nations Decade of Healthy Ageing (2021–2030) was proclaimed to in-crease the awareness on how healthy ageing is related with the following: healthy lifestyles; health literacy; integrated care; primary health services adapted to an older population; access to long-term care; opportunities for active participation; social participation; adequate levels of public expenditure in care and health [1].

If the decade’s vision is challenging for ‘normal’ ageing, it is critical to address ageing with neurocognitive minor or major disorders. According to the World Health Organization, in 2019, there were 55.2 million people in the world with dementia, a number which will reach 78 million by 2030 and 139 million by 2050. The prevalence of dementia worldwide is higher in the Western Pacific Region (20.1 million), European region (14.1 million), and Americas (10.3 million); it is also higher in older ages and women [2].

Neurocognitive disorders refer to cognitive impairment due to brain changes (e.g., memory, speech, perception, attention problems), and it differ from psychiatric disorders, chronic diseases or a lifestyle outcome. It also differs from age-associated cognitive decline that does not classify as disease and may configure a pre-morbid stage that progresses or not to dementia [3,4,5,6].

Recent reviews identified neurobiological markers for cognitive impairment in patients with psychiatric disorders. Biomarkers are not isolated indicators and should be linked with clinical criteria. The authors found pathogenetic factors for cognitive impairment in mental illness and social determinants for epigenetic mechanisms leading to mental illness [7,8]. Similar conclusions were addressed in recent studies about association between chronic disease, namely diabetes, and pathogenic and epigenetic factors of neurocognitive disorders [9,10].

These disorders are firstly noticed by the individual and the family, or by a family doctor or general practitioner (GP) in primary care regular appointments. Usually, a GP’s diagnosis should include clinical history (e.g., thyroid disease, malignancy, Vitamin B12 or other nutritional disorders, psychiatric diseases, strokes), and laboratorial exams (e.g., blood analysis, computed tomography, magnetic resonance imaging of the brain). After excluding other causes of probable dementia, the diagnosis should continue by screening for depression, doing cognitive assessment, namely, by using neuropsychological tests [11,12].

The World Alzheimer Report 2021 estimated that 75% of people with dementia do not have a regular diagnosis. The report also identified other concerns: lack of trained clinicians; fear of diagnosis and its cost; access to specialized diagnostic tests; beliefs about diagnosis utility; self-testing online and home tests; and medical appointments’ delays because COVID-19; new diagnosis tools (e.g., blood test) [13].

Primary care doctors in USA identified barriers to conduct a good diagnosis of neurocognitive disorders (e.g., Alzheimer’s disease), such as the following: a specific diagnosis not being essential as there are no adjusted treatments; lack of clarity regarding what to do with a dementia diagnosis; limited time; undervaluing of the importance of assessment and diagnosis; other patients’ problems taking precedence over cognitive problems; a lack of concrete guidelines or cutoff for screening dementia [14,15].

Moreover, in the older population, neurocognitive disorders diagnosis could be confused with normal ageing or other diseases, or it could be ignored because of ageism. This difficulty increases when older people are less educated, belong to lower social conditions or to non-normative groups (e.g., gender, ethnicity, location, class, income, education, social participation, religion, nutrition) [16,17,18]. Finally, ethical issues related to the disclosure of diagnosis in advances ages [19] may also appear as a barrier to diagnosis.

## 2. BRAINCODE in SHAPES for a Digital Diagnosis of Cognitive Impairment

Among technological and digital solutions for brain diagnosis, electroencephalography (EEG) medical devices are used by doctors to monitor brain activity and neurocognitive disorders [14,15,16]. This has the potential for differential diagnosis and biomarkers to distinguish different neurological and mental diseases, using brain markers for a specific neurocognitive disorder or another brain activity [20,21,22,23,24,25,26,27].

There are some challenges in the usage of EEG recordings for the characterization of pathological brain state: biological factors (e.g., medicine), environmental interferences (e.g., line noise); inter- and intra-subject variability of the extracted markers, e.g., Alpha, a brain rhythm that allows recognizing a brain disfunction, differs from individual to individual, and from different brain states in the same individual. Therefore, the EEG interpretation requires “an interprofessional team approach, including physicians, nurses, and mid-level providers, correctly trained” [28] (p. 10).

Starlab has been developing a digital technology for improving diagnosis of brain activity based on the EEG device called ENOBIO, which is currently commercialized by its linked company Neuroelectrics^®^. ENOBIO is a medical certified EEG device (CE, FDA) that together with NIC Desktop Software Platform provide a wireless, easy-to-use, and cost-effective brain activity measuring system. This technology has made it possible to develop BRAINCODE, a digital solution to assess cognitive impairment in older people. BRAINCODE aims to distinguish a normal from a pathologic brain condition, based in an EEG biomarkers evaluation at subject individual level. BRAINCODE includes not only a device (hardware/software), but also an EEG Data Driven Report methodology that works in three phases/levels: (i) EEG data acquisition; (ii) EEG feature extraction; (iii) BRAIN-CODE Report with the results (Figure 1).

Traditional electroencephalogram (EEG) analysis techniques focus on the spectral analysis EEG signals. Time-dependent signals are decomposed into a sum of pure frequency components using the Fourier transformation, which is grouped in bands Delta, Theta, Alpha, Beta and Gamma. BRAINCODE is based on EEG features corresponding to band power ratios associated with cognitive decline. These indices have been exploratory up to the moment since they depend on the following: the availability of large datasets; standardized protocols, montages, analysis techniques; validation in clinical trials. Indices were calculated in a reference population for the first time, and they need to be validated as normative values, indicating the level of cognitive impairment when applicable.

SHAPES Project funded BRAINCODE to be integrated in the SHAPES Platform. SHAPES will be a large-scale, EU-standardized open platform that integrates technological, organizational, clinical, educational, and societal solutions. This way increases long-term healthy and active ageing, the maintenance of a high-quality standard of life, and promote Health and Care (H&C) systems sustainability and networks with communities. SHAPES integration will have to be preceded by a SHAPES pan-European and large-scale pilot campaign with respect to their impact to improve health, wellbeing, in-dependence, and autonomy of older individuals, while enhancing the long-term sustain-ability of H&C systems in Europe [29].

## 3. Pilot Study: Purposes, Recruitment, Case–Control, Ethics, and Validation

The purpose of BRAINCODE’s pilot is to evaluate this technology’s efficacy in terms of correct differentiation of ‘Normal’ ageing, mild cognitive impairment, and dementia. It is also a purpose to provide guidelines about an adequate methodology, protocol, recruitment criteria, outcomes and its indicators to assess the BRAINCODE’s efficacy (Figure 2).

### 3.1. Recruitment: Criteria and Process

The participants will be doctors who will use BRAINCODE, and older adults who will be diagnosed. Older adults will be divided in two groups: participants with cognitive impairment (experimental group) and participants without cognitive impairment (control group). The inclusion and exclusion criteria are presented in Table 1.

These criteria must ensure that participants from the experimental group have a robust medical diagnosis of cognitive impairment; at the same time, participants from the control group cannot have any complains about cognitive functions and psychiatric disorders in the last 6 months. This is important to reduce the bias on BRAINCODE validation, namely confusion between cognitive impairment and psychiatric disorders.

The history of previous psychiatric disease is thought to be associated with subsequent episodes of depression and cognitive impairment which could bias clinical judgment [3,4,5,6,7,8], and even brain functioning, introducing spurious variables in that first phase of the validation process of BRAINCODE.

The idea to recruit participants older than 50 years and not only older ones aims to address on one hand the rise in younger people with dementia, and on the other to allow the future use of BRAINCODE as a prevention tool to deploy early customized intervention for MCI [4].

A list of potential doctors from neurology, psychiatric or geriatric services, settled on healthcare organization with an ethical committee, will be invited by email and phone calls (maximum four times for a month). There will be a meeting (presential, online, phone call) with those that accept to participate, to explain the pilot study and protocol.

The older adults will be recruited by doctors during the regular medical appointments. Considering recruitment criteria, older adults will be divided into two groups: a group of ‘patients’ with cognitive impairment (experimental group); a group comprising a respective accompanying person or ‘caregiver’ who has no subjective cognitive complaints (control group).

### 3.2. Protocol and Ethics

The pilot study will adopt a case–control methodology to compare BRAINCODE’s performance with medical diagnosis and neuropsychologic tests. The study will be developed in two pilot sites: Portugal, in a neurology and/or psychiatric service based on a central hospital of the National Health System (that works in collaboration with the University of Porto); Ireland, in a Centre for Gerontology and Rehabilitation of University College Cork.

During the medical appointment, doctors will invite ‘patients’ and ‘caregivers’ to be assessed by BRAINCODE and neuropsychologic tests. While the accompanying person or ‘caregiver’ will be selected by denying having subjective cognitive complains, patients will follow the regular clinical procedures before the experimental phase.

These procedures will be blinded to the other researchers. There will be three independent and non-communicable teams: doctors responsible for recruitment and regular clinical diagnosis; psycho-gerontologists responsible for administering neuropsychologic tests; Starlab’s team, which is responsible for BRAINCODE screening and reports; University of Porto and University College Cork, which are responsible for validating results. There will not be data/information exchange between teams until both groups of older adults have been fully evaluated, and University of Porto and University College Cork have validated the results.

The protocol comprises four assessment tools. The clinical diagnosis corresponds to all the usual procedures that doctors use to deliver a medical diagnosis of neurocognitive disorder or mild cognitive impairment. The list of exams depends on each situation but often includes clinical history, laboratorial exams, psychiatric and neuropsychological examinations.

The BRAINCODE scan is an EEG recording for a 15 min protocol of interleaved eyes open and closed in resting state using an ENOBIO 32 Neuroelectrics device. The montage consists of 32 electrodes filled with gel. It is important to note that dry electrodes could also be used in the future (e.g., for EEG home applications), which reduce montage time and increase the ease of use for participant’s (since the hair is not wet). Nevertheless, this pilot will always use gel to obtain signals with the standard usage to obtain a reference evaluation and validation. After the signals are recorded, BRAINCODE will generate the final report with results. Clinicians performing the EEG exam will be previously trained remotely in a synchronous manner (since asynchronous has a lower reliability and sensitivity) via video conferencing.

The MoCA is a cognitive test to differentiate normal from pathological cognitive decline, and it is being adopted worldwide by healthcare professionals due to its higher sensitivity compared with Mini-Mental State Exam. The maximum score is 30 points; the authors suggested a cut off score of 26. This test assesses cognitive functions and abilities such as memory, attention, language, abstraction, and orientation; the test is adapted to 12 years of formal education. The test administration takes around 10 min by a professional [30,31,32,33].

The QMCI is a quick neuropsychological test (3–5 min) for detecting cognitive impairment levels (MCI or dementia). The test is generally used as a complement to clinical examinations and evaluates orientation, words registration (5 points), clock drawing, one-minute delays recall, verbal category fluency and logical memory by means of a verbal recall of a short story. The total score is 100 points and the optimal established cut-off for cognitive impairment is 62 after correction of education and culture [34,35,36,37].

This pilot study will ensure ethical and data protection procedures, since it will enroll older people in a real medical context and since it is framed on a SHAPES project. There will be the following: (i) an ethical approval from participants’ healthcare organizations and from universities engaged in the study; (ii) a consent form that respects the fundamental rights, the biomedical ethics and ethics of care, the Declaration of Helsinki, and the Oviedo Convention; (iii) a clear explanation of the project and the free and informed consent form; (iv) a signed informed consent by each participant; (v) a Data Protection Impact Assessment, a personal data processing description, and data security and a risk assessment documents; (vi) a signed data processing agreement and a data sharing agreement.

### 3.3. Validation

This case–control pilot aims to validate the BRAINCODE’s efficacy, facing the challenges related to neurocognitive disorders diagnosis in older adults. The main objectives are to verify if BRAINCODE confirms clinical diagnosis, differentiates cognitive capacity between experimental and control groups, confirms cognitive impairment measured by QMCI and MoCA, and differentiates between MCI and dementia. Cut-off points are selected on a comparative study between QMCI and MoCA [38] (Figure 3).

### 3.4. Limitations

This pilot study previews a small number of participants using an experimental case–control design not controlling for known variables associated with cognitive impairment, namely, education and sex, and using a wide range of ages. The control group (accompanying person) will be classified as not having cognitive impairment based only on a question about having or not subjective memory complains. The pilot put together participants from two different countries which may introduce additional cultural variance.

## 4. Conclusions

Currently, BRAINCODE identifies a brain condition by an EEG biomarkers evaluation at individual level. This is performed by comparing individual EEG results with normative values calculated by scores from an older population (normal/pathological) available on scientific literature or data bases. Scores are based on EEG markers related to ageing and cognitive decline and are displayed in a Report in a way to help clinicians interpret the results.

The success of BRAINCODE’s validation in a real medical context will lead to an increase in clinicians’ confidence on this technology to complement traditional medical diagnostics in the long run. The decision makers will be prone to adopt it to raise diagnosis accuracy and accessibility. Researchers and Starlab will have information about a case–control pilot study that could be supported, a robustness validation study and usability evaluation.

BRAINCODE will represent a disruptive opportunity to reduce sub-diagnosis and/or late diagnosis, and negative impacts of neurocognitive disorders in an older population. Since it represents an opportunity to deliverable an early, extensive, accurate, and cost-effective clinical diagnosis of neurocognitive disorders, this technology can complement conventional diagnostic methods and surpass certain limitations such as the dependency of patients on language and question comprehension in certain cognitive tests, extensive test times and need of expert knowledge for issuing the diagnosis. Additionally, the technology could increase neuroprotective interventions to prolong healthy brain lifetime; its portability and easiness to use also represents an opportunity for health professionals, in isolated territories and primary care services, without easy and direct access to medical experts and EEG equipment, being able to deliver a diagnosis and refer patients for further assistance.

## Figures and Tables

**Figure 1 ijerph-19-05768-f001:**
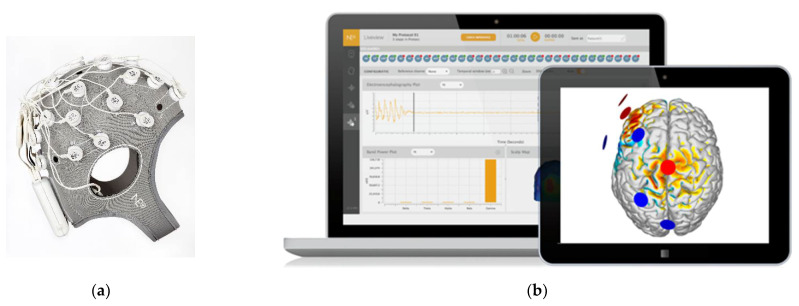
BRAINCODE’s hardware and software: (**a**) medical certified EEG device ENOBIO; (**b**) NIC Desktop Software Platform (Software creator: Neuroelectrics, Barcelona, Spain. Version number: 2.1.0.).

**Figure 2 ijerph-19-05768-f002:**
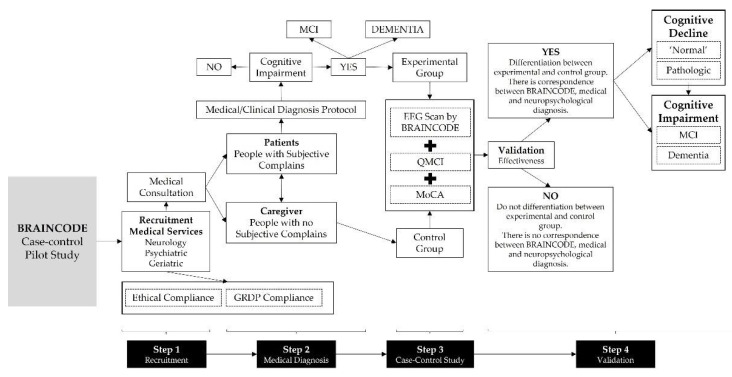
Pilot study design.

**Figure 3 ijerph-19-05768-f003:**
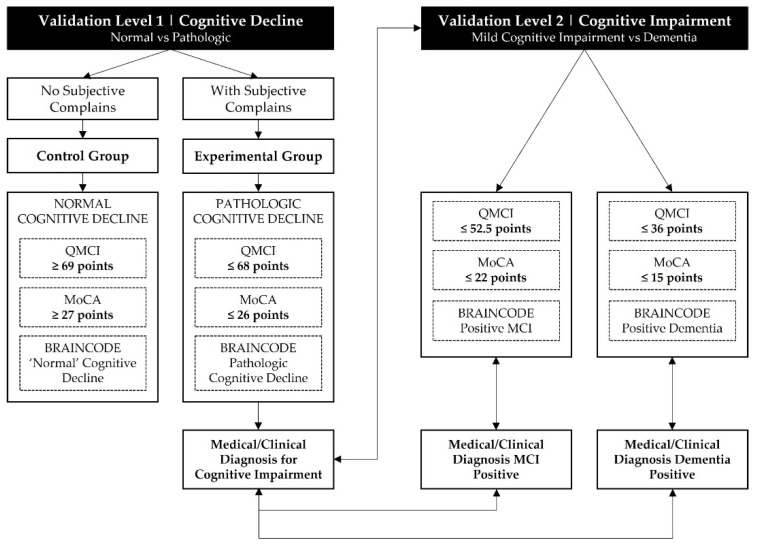
Validation process.

**Table 1 ijerph-19-05768-t001:** Recruitment criteria by participant’s types.

**Participants’ Categories**	**Inclusion Criteria**	**Exclusion Criteria**
Patients (Experimental Group) From 10 to 20 participants	Older Adults (≥50 years old). Having a diagnosis of minor or major neurodegenerative impairment (MCI or dementia). Subjective complaint of memory loss by the patient or the family during the last 6 months. Being able to attend a medical appointment. Provide an ethical consent form, and data protection consent form.	Having other severe medical conditions (e.g., stroke, epilepsy, meningoencephalitis, brain tumor, severe concussion, multiple sclerosis). Having history of previous psychiatric disease within the last 10 years (bipolar disorder, posttraumatic stress disorder, severe depression, psychosis, attempted suicide). Being a drug addict (e.g., alcohol, MDMA, amphetamines, cocaine, opiates, benzodiazepine, cannabis) Interrupts the research participation process.
Patients (Control Group) From 10 to 20 participants	Older Adults (≥50 years old). Without neurodegenerative disease diagnosed. Having no subjective memory complains. Provide an ethical consent form, and data protection consent form.	Having severe medical and or psychiatric conditions (e.g., stroke, epilepsy, meningoencephalitis, brain tumor, severe concussion, multiple sclerosis). Subjective complaint of memory loss during the last 6 months.
Clinicians From 1 to 5 participants	Being a medical doctor (e.g., neurologist, psychiatric, geriatric). Working in a healthcare organization with ethics committee. Obtain ethical approvements from their own healthcare organization. Use to work with neurodegenerative disease. Have regular practice with patients who are ≥50 years old.	Do not sign data protection agreements. Null experience with EEG analysis and/or recording.

## Data Availability

There are no data available yet.

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
