# Peer review of "BRAINCODE for Cognitive Impairment Diagnosis in Older Adults: Designing a Case–Control Pilot Study"

_ijerph, 2022, doi:10.3390/ijerph19095768_

Round 1

Reviewer 1 Report

The cognitive impairment diagnosis in older adults is an issue, which within the aging of the society, will gain more attention. The limited number of methods providing sufficient assessing tools, undoubtedly leads to the necessity for search for more optimal tools. Some issue should be changed before further processing:

1. My major concern is the poor background of the project mentioned in the introduction. e.g. Authors mention comorbidities as diabetes, however do not bring up the issue of glycemic variability and prediabetes and their possible impact on cognitive abilities (Ref.)

  1. Cognitive function among older adults with diabetes and prediabetes, NHANES 2011-2014. Diabetes Res Clin Pract. 2021 Aug;178:108939. doi: 10.1016/j.diabres.2021.108939. Epub 2021 Jul 3. PMID: 34229005; PMCID: PMC8429258.
  2. The Rate of Decrease in Brain Perfusion in Progressive Supranuclear Palsy and Corticobasal Syndrome May Be Impacted by Glycemic Variability-A Pilot Study. Front Neurol. 2021 Nov 8;12:767480. doi: 10.3389/fneur.2021.767480. PMID: 34819913; PMCID: PMC8606811.

2. The limitations of this project should be clearly stated.

3. In table 1 the term "other sever medical conditions" is not sufficiently precise.

Author Response

Point 1: My major concern is the poor background of the project mentioned in the introduction. e.g. Authors mention comorbidities as diabetes, however do not bring up the issue of glycemic variability and prediabetes and their possible impact on cognitive abilities (Ref.) 1. Cognitive function among older adults with diabetes and prediabetes, NHANES 2011-2014. Diabetes Res Clin Pract.2021 Aug;178:108939. doi: 10.1016/j.diabres.2021.108939.Epub 2021 Jul 3. PMID: 34229005; PMCID: PMC8429258. 2. The Rate of Decrease in Brain Perfusion in Progressive Supranuclear Palsy and Corticobasal Syndrome May Be Impacted by Glycemic Variability - A Pilot Study. Front Neurol.2021 Nov 8;12:767480. doi: 10.3389/fneur.2021.767480.PMID: 34819913; PMCID: PMC8606811.

Response 1: The authors are grateful for the comment that were relevant to review the article, namely to include new references. Please check the corrections (pp.1-2): “Neurocognitive disorders refer to cognitive impairment due to brain changes (e.g., memory, speech, perception, attention problems), and it differ from psychiatric disorders, chronic diseases or a lifestyle outcome. It also differs from age associated cognitive de-cline that does not classify as disease and may configure a pre-morbid stage, that pro-gresses or not to dementia [3-6]. Recent reviews identified neurobiological markers for cognitive impairment in pa-tients with psychiatric disorders. Biomarkers are not isolated indicators and should be linked with clinical criteria. Authors found pathogenetic factors for cognitive impairment in mental illness and social determinants for epigenetic mechanisms leading to mental illness [7,8]. Similar conclusions were addressed in recent studies about association be-tween chronic disease, namely diabetes, and pathogenic and epigenetic factors of neu-rocognitive disorders [9,10]”.

Point 2: The limitations of this project should be clearly stated.

Response 2: Thank you for this observation. We include a new point 3.4, (p.7): “This pilot study previews a small number of participants using an experimental a case control design not controlling for known variables associated with cognitive impair-ment namely education and sex, using a wide range of ages. The control group (accom-panying person) will be classified as not having cognitive impairment based only in a question about having or not subjective memory complains. The pilot put together partic-ipants from two different countries which may introduce additional cultural variance”.

Point 3: In table 1 the term "other sever medical conditions" is not sufficiently precise.

Response 3: Thank you for this comment. the authors do not want to restrict the list of “medical conditions” that will be consider by the doctor that will recruit and make the diagnosis of participants but include an extended explanation about recruitement criteria (p.4): “These criteria must ensure that participants from experimental group have a robust medical diagnosis of cognitive impairment; at the same time, participants from control group  do not have any complains about cognitive functions and psychiatric disorders, in the last 6 months. This is important to reduce the bias on BRAINCODE validation, namely confusion between cognitive impairment and psychiatric disorders. The history of previous psychiatric disease is thought to be associated with subse-quent episodes of depression and cognitive impairment which could bias clinical judg-ment [3-8], and even brain functioning affecting introducing spurious variables in that first phase of the validation process of BRAINCODE. The idea to recruit participants older than 50 years and not only older ones aims to address on one hand the rise of younger people with dementia, and on the other to allow the future use of BRAINCODE as a prevention tool to deploy early customized interven-tion for MCI [4]”.

Reviewer 2 Report

This study includes grateful promise in cognitive impairment diagnosis. Some revises are requested to make the results more meaningful.

ï¼»Majorï¼½

There are few concrete explanations about how BRAINCODE for cognitive impairment diagnosis was superior to conventional diagnostic methods and what problems may arise. It is necessary to mention the reason why the BRAINCODE methods could not be used until now, such as unestablished technology and cost. Explaining these reasons will be useful as new technologies become more widespread. We request corrections and corrections to the above parts, but if it is determined that it is not necessary, please explain the reason as it is a previous study.

ï¼»Minorï¼½

1) Explain why this study defined older adults as over 50 years old. Many studies were defined older adults as over 65 years old.

2) In exclusion criteria, why the history of previous psychiatric disease within the last 10years?

Author Response

Point 1: There are few concrete explanations about how BRAINCODE forcognitive impairment diagnosis was superior to conventionaldiagnostic methods and what problems may arise. It isnecessary to mention the reason why the BRAINCODE methodscould not be used until now, such as unestablished technologyand cost. Explaining these reasons will be useful as newtechnologies become more widespread. We request corrections and corrections to the above parts, but if it is determined that it isnot necessary, please explain the reason as it is a previousstudy.

Response 1: The authors are grateful for the comment that were relevant to review the article, namely to include more information about BRAINCODE (p.3 and p.7):

“Traditional Electroencephalogram (EEG) analysis techniques focus on the spectral analysis EEG signals. Time-dependent signals are decomposed into a sum of pure fre-quency components using the Fourier transformation, which is grouped in bands Delta, Theta, Alpha, Beta and Gamma. BRAINCODE is based on EEG features corresponding to band power ratios associated with cognitive decline. These indices have been exploratory up to the moment since they depend on the availability of large datasets; standardized protocols, montages, analysis techniques; and validation in clinical trials. Indices were calculated in a reference population for the first time, and they need to be validated as normative values, indicating the level of cognitive impairment when applicable”

“this technology can complement conventional diagnostic methods and surpass certain limitations such as the dependency of patients to language and question comprehension in certain cognitive tests, extensive test times and need of expert knowledge for issuing the diagnosis”

Point 2: Explain why this study defined older adults as over 50 yearsold. Many studies were defined older adults as over 65 yearsold.

Response 2: The authors are grateful for this observation, and accordingly included further explanations (p.4): “The idea to recruit participants older than 50 years and not only older ones aims to address on one hand the rise of younger people with dementia, and on the other to allow the future use of BRAINCODE as a prevention tool to deploy early customized interven-tion for MCI [4]”.

“These criteria must ensure that participants from experimental group have a robust medical diagnosis of cognitive impairment; at the same time, participants from control group do not have any complains about cognitive functions and psychiatric disorders, in the last 6 months. This is important to reduce the bias on BRAINCODE validation, namely confusion between cognitive impairment and psychiatric disorders”.

Point 3: In exclusion criteria, why the history of previous psychiatric disease within the last 10 years?

Response 3: Thank you for the question. Although the amount of years chosen (10 years) may be considered arbitrary, the authors think that if during the previous 10 years there was no episode of psychiatric disease the probability of any condition to interfere with the actual diagnosis is small. However, considering the pretinence of the question, the authors included a new explanation (p.4): “The history of previous psychiatric disease is thought to be associated with subse-quent episodes of depression and cognitive impairment which could bias clinical judg-ment [3-8], and even brain functioning affecting introducing spurious variables in that first phase of the validation process of BRAINCODE”.

“These criteria must ensure that participants from experimental group have a robust medical diagnosis of cognitive impairment; at the same time, participants from control group do not have any complains about cognitive functions and psychiatric disorders, in the last 6 months. This is important to reduce the bias on BRAINCODE validation, namely confusion between cognitive impairment and psychiatric disorders”.

This comment was inline with the backgroud revision, (p.1): “Recent reviews identified neurobiological markers for cognitive impairment in pa-tients with psychiatric disorders. Biomarkers are not isolated indicators and should be linked with clinical criteria. Authors found pathogenetic factors for cognitive impairment in mental illness and social determinants for epigenetic mechanisms leading to mental illness [7,8]”.

Round 2

Reviewer 1 Report

I do not have further comments.